# Comparison of Electroencephalogram Power Spectrum Characteristics of Left and Right Dragon Boat Athletes after 1 km of Rowing

**DOI:** 10.3390/brainsci12121621

**Published:** 2022-11-25

**Authors:** Yan Zhang, Hongke Jiang, Wu Zhou, Yingying Cao, Changzhuan Shao, Jing Song, Aiping Chi

**Affiliations:** 1School of Sports, Shaanxi Normal University, Xi’an 710119, China; 2Physical Education Department, Shanghai Maritime University, Shanghai 200135, China

**Keywords:** dragon boat, EEG, power spectrum, hemispheric asymmetry

## Abstract

Purpose: This study aimed to detect differences in post-exercise brain activity between the left and right paddlers due to exercise by analyzing the resting-state electroencephalogram (EEG) power spectrum before and after exercise. Methods: Twenty-one right paddlers and twenty-two left paddlers completed a 1 km all-out test on a dragon boat ergometer, and their heart rate and exercise time were recorded. EEG signals were collected from superficial brain layers before and after exercise; then, the EEG power spectrum was extracted and compared in different frequency bands. In addition, the degree of lateralization in each brain region was assessed by the asymmetry index. Results: There was no significant difference in the power spectrum values and asymmetry indices between the left and right paddlers before rowing (*p* ˃ 0.05). However, after rowing, the left-paddlers group had significantly higher spectral power values in θ and α bands than the right-paddlers group (*p* < 0.05), and brain lateralization in both groups of athletes occurred mainly in the ipsilateral hemisphere of the frontal and central regions. Conclusion: The 1 km of rowing induced more brain activation in the left paddlers, and both left and right paddlers showed functional aggregation of hemispheric lateralization.

## 1. Introduction

The dragon boating contest is a traditional Chinese sport that has been around for over 2000 years. Every May, large dragon boat races are held in some parts of China, and the event was officially included in the 2007 Asian Games. The dragon boat is a team sport that requires paddlers to sit on either side of the boat and paddle in unison to propel the boat forward. The sport requires great skill and physical ability from the paddlers, especially the consistency of stroke rate, range of motion, strength and rhythm between paddlers, which plays a key role in the direction and speed of the boat. It is well known that cortical control of voluntary movements is crossed, with one side of the motor cortex dominating the activity of the other side of the body [1], which means that the areas of proprioceptive projection to the cerebral cortex are contralateral [2]. In addition, the somatosensory system is bilaterally structured and, when adopted to the right and left sides of the body, the two cerebral hemispheres would produce different electrophysiological stimulation [3]. Therefore, the dragon boat sport may have different effects on the neurological activity in the brains of the left and right paddlers.

Asymmetry of the brain is a unique human trait that is further developed and lateralized through specialization in labor or exercise [4]. Several studies have confirmed that unimanual movement tasks activate hemispheric asymmetries in the motor cortex [5,6]. In a test of unilateral hand muscle-triggered motor evoked potentials (MEPs), Muellbacher et al. [7] observed changes in motor cortex activation on the same side as the task hand. It has been found through functional magnetic resonance imaging (fMRI) studies that, when an individual makes unilateral finger movements, there is typically activation of the motor cortex on the same side [8]. However, the hemispheric asymmetrical nature of these activations is unclear. It could be a subthreshold mirror activity propagating from the predominantly active contralateral motor cortex to the ipsilateral motor cortex, or it could be caused by activation of ipsilateral or bilaterally distributed motor pathways [9]. It is well known that, for athletes involved in unilateral sports, such as throwing, tennis, badminton and table tennis, there are asymmetries in the brain regions responsible for motor command delivery and sensory processing [10,11,12]. For the dragon boat sport, the left and right paddlers are fixed and play a specialized sporting role. Due to this inherent dragon boat sport characteristic, asymmetries in their post-exercise EEG signals might be present. However, evidence for this hypothesis is lacking.

Studies investigating the differences between dominant hands often find significant variations in performance on exercise tasks. Taddei et al. [13] found that left-handed fencers had faster visual responses than right-handed fencers. McLean and Ciurczak [14] found that left-handed baseball players had better mean batting performance. A study of asymmetry in sport also showed that left-handed players have a great strategic advantage over right-handed players in tennis [15]. One potential origin of these differences could be due to the hemispheric structure of the brain and hemispheric lateralization in movement control processes. Studies have shown that the corpus callosum is often larger in left-handedness [16], which may be a sign of better connectivity between the cerebral hemispheres [17]. In another study of transcranial direct current stimulation (tDCS), it was confirmed that the lateralization of left vestibular cortical function was more pronounced in the left-handed [18]. In conclusion, the above studies confirm that handedness affects cortical activity and suggest that left-handed people have higher cortical activation.

The left and right paddlers in the dragon boat perform symmetric sporting technical movements, and the hemispheric asymmetry of the athletes’ brains may vary significantly. To explore this issue, this study selected left and right paddlers in the dragon boat and used a dragon boat ergometer to conduct a 1 km rowing test, and examined the differences in brain activation between the two types of dragon boat paddlers by collecting their resting state EEG signals before and after the 1 km rowing test. In this paper, we use the Asymmetry Index for comparison to further analyze the areas where there are differences in both hemisphere activation between left and right paddlers, providing a reference point for understanding the differences in brain activity caused by exercise in unilateral-sport athletes.

## 2. Materials and Methods

### 2.1. Participants

Forty-five male elite athletes (age range: 18~25 years) were recruited, all of whom have been competing in the national tournament. Two subjects (2 left paddlers) dropped out during the course of the study due to physical discomfort. The final participants consisted of 43 athletes (21 left paddlers and 22 right paddlers). To exclude possible interfering factors from the handedness, all selected athletes were right-handed (measured by the Chinese translated version of Edinburgh inventory [19]). Other physical illnesses and dysplasia were excluded, and among the athletes there was no history of major injuries. The training age of the athletes was more than 3 years, and their training times were more than 7 h per week. This research was obtained from the Ethics Board of Shaanxi Normal University (No. 202116012) in accordance with the Declaration of Helsinki, and all subjects signed the informed consent paper after knowing the procedure and purpose of this experiment. Basic information on subjects was shown in Table 1.

### 2.2. Experimental Protocol

The test was conducted on the D1-M dragon boat ergometer (KayakPro, Miami, FL, USA) indoors. All subjects were informed of the need to comply with the following requirements prior to testing: do not perform high-intensity exercise within 48 h before the test, avoid any stimulants such as tea and coffee, etc. for at least 5 h before the test, do not eat or drink for at least 3 h prior to the test. The trial time was 9:00 a.m.–11:00 a.m. in the morning. Exercise method: The participants warmed up for 10 min with low-intensity rowing, followed by an all-out 1 km rowing exercise. The subject was instructed to grab the ergometer handle and posited themselves in a fully compressed ready position. When the subject signaled that he was ready, the researcher gave the command “READY, GO”. At that time, the subject pulled as hard as possible, trying to achieve full extension with the stroke. Throughout the testing process, the subjects wore a heart rate monitor (GT9-X, ActiLife, Pensacola, FL, USA) to collect heart rate data. The resting heart rate (HRrest) and the EEG data in post-exercise and pre-exercise were collected while the subjects were quiet and sitting. The acquisition time was at least 6 min. The experimental protocol was shown in Figure 1.

### 2.3. Collection and Preprocessing of EEG Data

EEG data were recorded for 6 min for all subjects using the Neuroscan acquisition amplifier (Neuroscan Inc., El Paso, TX, USA) from 32 scalp locations (FP1, FP2, F7, F3, FZ, F4, F8, T3, T5, C3, CZ, C4, T4, T6, M1, P3, PZ, P4, M2, O1, O2, TP8, TP7, OZ, CP4, FC3, FT8, CP3, FT7, CPZ, FC4, FCZ), according to the international 10/20 system. Participants were instructed to wash their heads before and after exercise (disable hair care products such as conditioners), and then recording the EEG data, which reduce interference of EEG signals for scalp dirt. During EEG data acquisition, participants remained in a relaxed awake state with their eyes closed and sat in an upright position, avoiding conditions that could cause artifacts such as swallowing, eye movements, and frowning. The examination room was dimly-lit and soundproof. The sampling rate of EEG data was recorded at 1024 Hz, with mastoids M1 and M2 serving as references. Each channel impedance was maintained below 5 kΩ and the acquisition time was at least 6 min. Off-line analysis of raw EEG signals was performed using the EEGLAB toolbox in MATLAB R2013b software (MathWorks, Natick, MA, USA). Firstly, the EEG data were down-sampled to 500 Hz and bandpass filtered with frequency between 1–45 Hz and notch filtering with frequency between 48–52 Hz. Use Independent Component Analysis (ICA) and manual artifact judgment to eliminate of eye-movement and muscular artifacts. Data were standardized and segmented into 2 s epochs and re-referenced by whole brain average reference value [20].

### 2.4. EEG Power Spectrum Analysis

The spectrum of the preprocessed EEG data was calculated using the Pwelch method function of MATLAB R2013b. The Fast Fourier transform was conducted to estimate the EEG power spectrum density of δ (1–4 Hz), θ (4–8 Hz), α (8–13 Hz), and β (14–30 Hz) frequency. Let *X*(*n*) be an EEG spectrum, and then each segment power *I_k_* and the whole segment power *P*(*f_n_*) were calculated through Equation (1) and Equation (3).

Sampling point *n* = 0, 1, …, *N*. *N* is the total length of the data. Divide into *X_1_ (n)*, *X_2_* (*n*), ..., *X_k_* (*n*), ..., *X_k_* (*n*) a total of *K* data segments covering the entire data (*K* − 1) × *D* + *L* = *N*, where *D* is the step length of each window function move, *L* is the length of each data segment, *k* = 1, 2, ..., *K*. Let the window function be *W*(*n*) with sampling points *n* = 0, 1, ..., *L* − 1 and data segments denoted as *X*_1_ (*n*)*W*(*n*), ..., *X_k_* (*n*)*W*(*n*). *h* represents EEG rhythms, (e.g., δ, θ, α, β), and *E*_total_ represents the total power of all EEG rhythms.

Equation (1):(1)$$I_{k}\left(f_{n}\right)=\frac{1}{UL}{\left|\overset{L-1}{\underset{n=0}{\sum }}X_{k}\left(n\right)W\left(n\right)\exp\left(\frac{-i2\pi fn}{L}\right)\right|}^{2}$$

*U* is as in Equation (2):(2)$$U=\frac{1}{L}\sum_{n=0}^{L-1}W{\left(n\right)}^{2}$$

Equation (3):(3)$$P\left(f_{n}\right)=\frac{1}{K}\sum_{k-1}^{K}I_{K}\left(f_{n}\right)=\frac{1}{ULK}\sum_{k-1}^{k}{\left|\overset{L-1}{\underset{n=0}{\sum }}X_{k}\left(n\right)W\left(n\right)e^{\frac{-i2\pi fn}{L}}\right|}^{2}$$
$$R\left(h\right)=\frac{P\left(f_{n}\right)}{E_{total}}$$

The current study assessed EEG power asymmetry in athletes between different paddlers. To identify the differences in activation of brain regions, it divided the whole brain region into five brain functional areas: frontal (FP1–F7, FP2–F8), central (F3–C3, F4–C4), parietal (C3–P7, C4–P8), temporal (F7–P7, F8–P8), and occipital (P3–O1, P4–O2) regions. Asymmetry indices (AIs) were generated for homologous right-left electrode pairs using the following standard calculation: [(R − L)/(R + L) × 100%], where R and L were the specific frequency band power of the corresponding electrode [21,22]. Frequency bands were defined as follows: δ (1~4 Hz), θ (4~8 Hz), α (8–13 Hz), β (14~30 Hz).

### 2.5. Statistical Analysis

SPSS 26.0 (version 22.0; SPSS Inc., Chicago, IL, USA) was used for the statistical analysis, and all data were expressed as mean ± standard deviation ($\bar{x}$ ± *s*). Use Kolmogorov–Smirnov to test the normal distribution of each group of data which from the two paddlers groups were compared by mixed design analysis of variance (ANOVA). Post-hoc comparisons were performed by the Independent Samples *t*-tests and Bonferroni multiple comparison test. All ANOVAs were performed using Mauchly’s test of sphericity to determine whether it was conformed; if there was a violation, the Greenhouse–Geisser adjustment would be adopted. Simple effect analysis was performed for data with interaction effects, and *p* < 0.05 was considered statistically significant. A two-way mixed design ANOVA was carried out for the EEG power spectrum value with a number of correct items as the dependent variables, paddlers (right, left) as a between-subject factor, and the frequency bands (δ, θ, α, β) as a within-subject factor. To test the AI for each brain region, a two-way mixed design ANOVA was performed with paddlers (right, left) as a between-subject factor and the electrode pairs as a within-subject factor. When there were less than two electrode pairs (occipital area), one-way ANOVA was used.

## 3. Results

### 3.1. Comparison of Athletes’ Post-Exercise Performance

The results of comparing the athletic performance of the two groups of paddlers are shown in Table 2.

In order to better assess the exercise performance, four indicators (HR peak, Percentage of HR max (%), Physiological load index, Rating of Perceived Exertion (RPE), and Exercise duration) were used as evaluation criteria in this experiment (Table 2).

In the process of the experiment, the HR peak of both the left paddler group and the right paddler group were greater than 190 b/min, and the Percentage of HR max was 95% and 96%, which were in the range of super-intense exercise. The Physiological load index showed that both the left paddler group and the right paddler group were greater than 1.6, which was far beyond the high degree. In addition, all subjects in both groups achieved an RPE level of 18–20, and there was no significant difference in exercise performance data between the two groups by independent samples *t*-test (*p* = 0.411 > 0.05).

### 3.2. Comparison of EEG Power Spectrum Values by Frequency Bands for Athletes

2(paddlers) × 4(frequency bands) mixed design analysis of variance was conducted for the power spectrum value. The results revealed no significant interaction effects between paddlers and the frequency band before rowing [*F*_(1, 41)_ = 2.211, *p* = 0.130, *η*^2^ = 0.091], and the main effects of the two factors were not significant. There were interaction effects between paddlers and frequency band after rowing [*F*_(1, 41)_ = 5.422, *MSE* = 0.143, *p* < 0.05, *η*^2^ = 0.198]. Simple effect analysis found that, in the θ and α bands, there was a difference in the power spectrum at different paddlers [*F*_(1, 41)_ = 16.637/23.069, *p* < 0.01]. The power of θ in the left-paddlers group [*M* = 1.457, 95%CL (1.261, 1.653)] was significantly larger than that in the right-paddlers group [*M* = 0.913, 95%CL (0.717, 1.109)], *p* < 0.001. The power of α in the left-paddlers group [*M* = 1.315, 95%CL (1.200, 1.430)] was significantly larger than that in the right-paddlers group [*M* = 0.939, 95%CL (0.825, 1.054)], *p* < 0.001. The results are shown in Figure 2.

### 3.3. Comparison of Asymmetry Indices between Two Groups of Dragon Boat Athletes

The AI was calculated for each brain region using the power values of each EEG channel in θ and α bands, where Fp2–Fp1 and F4–F3 were compared for frontal left-right asymmetry, FC4–FC3, C4–C3 and CP4–CP3 compared for central left-right asymmetry, FT8–FT7, T8–T7, and TP8–TP7 compared for temporal left-right asymmetry, P8–P7 and P4–P3 compared for left-right asymmetry in the parietal lobe, and O2–O1 compared for left-right asymmetry in the occipital lobe. The results of the brain area AIs in the θ and α bands after rowing for the left-paddlers group and right-paddlers group are shown in Figure 3 and Figure 4.

2(paddlers) × 3(electrode pairs) mixed design analysis of variance was conducted for the frontal asymmetry. The results revealed interaction effects between the paddlers and electrode pairs in the θ band [*F*_(2, 82)_ = 4.789, *p* < 0.05, *η*^2^ = 0.179]. Simple effect analysis found that the AI of F4 − F3 was a significant difference between paddlers [*F*_(1, 41)_ = 5.504, *p* < 0.05]. The AI of F4 − F3 in the right-paddlers group [*M* = 0.066, 95%CL (−0.013, 0.145)] was significantly higher than that of the left-paddlers group [*M* = −0.060, 95%CL (−0.139, 0.019)], *p* < 0.05. In the α band, interaction effects between paddlers and electrode pairs [*F*_(2, 82)_ = 8.204, *p* < 0.01, *η*^2^ = 0.272]. The simple effect analysis indicated that the AIs of two sets of electrode pairs were significant differences between paddlers [*F*_(1, 41)_ = 6.724/5.134, *p* < 0.05]. The AI of F8 − F7 in the left-paddlers group [*M* = −0.076, 95%CL (−0.216, 0.064)] was significantly decreased compared to that in the right-paddlers group [*M* = 0.172, 95%CL (0.032, 0.311)], *p* < 0.05. The AI of F4 − F3 in the left-paddlers group [*M* = −0.088, 95%CL (−0.232, 0.055)] was significantly decreased compared to that in the right-paddlers group [*M* = 0.133, 95%CL (−0.010, 0.276)], *p* < 0.05.

2(paddlers) × 3(electrode pairs) mixed design analysis of variance was conducted for the central asymmetry. The analysis announced significant interaction effects between the paddlers and electrode pairs in the θ band [*F*_(2, 44)_ = 5.843, *p* < 0.01, *η*^2^ = 0.210]. Simple effects analysis revealed significant differences between the two sets of electrode pairs AIs across the paddlers [*F*_(1, 41)_ = 5.466/5.395, *p* < 0.05]. The AI of FC4 − FC3 in the left-paddlers group [*M* = −0.077, 95%CL (−0.175, 0.022)] was significantly decreased compared to that in the right-paddlers group [*M* = 0.080, 95%CL (−0.018, 0.179)], *p* < 0.05. The AI of C4–C3 in the left-paddlers group [*M* = −0.074, 95%CL (−0.173, 0.025)] was significantly decreased compared to that in the right-paddlers group [*M* = 0.082, 95%CL (−0.017, 0.181)], *p* < 0.05. In the α band, a significant main effect of paddlers for the central asymmetry [*F*_(1, 41)_ = 5.335, *p* < 0.05, *MSE* = 0.103, *η*^2^ = 0.195]. The post-hoc comparison conducted that the AI in the right-paddlers group [*M* = 0.143, *SE* = 0.053] was significantly higher than that of the left-paddlers [*M* = −0.031, *SE* = 0.053]. Further analysis conducted a significant interaction between the paddlers and electrode pairs [*F*_(2, 82)_ = 4.580, *p* < 0.05, *η*^2^ = 0.172]. Simple effects analysis revealed significant differences between the two sets of electrode pairs AIs across the paddlers [*F*_(1, 41)_ = 8.380/8.176, *p* < 0.01]. The AI of FC4 − FC3 in the left-paddlers group [*M* = −0.082, 95%CL (−0.201, 0.037)] was significantly decreased compared to that in the right-paddlers group [*M* = 0.152, 95%CL (0.034, 0.271)], *p* < 0.01. The AI of C4 − C3 in the left-paddlers group [*M* = −0.092, 95%CL (−0.226, 0.042)] was significantly decreased compared to that in the right-paddlers group [*M* = 0.169, 95%CL (−0.035, 0.303)], *p* < 0.01.

2(paddlers) × 3(electrode pairs) mixed design analysis of variance indicated for the temporal asymmetry. The analysis revealed that, in the θ band, there was a significant interaction effects between paddlers and electrode pair [*F*_(2, 82)_ = 3.893, *p* < 0.05, *η*^2^ = 0.149]. Simple main effect analysis conducted that the AI of FT8–FT7 was a significant difference between paddlers [*F*_(1, 41)_ = 4.865, *p* < 0.05]. The right-paddlers group showed a significantly positive AI of FT8–FT7 [*M* = 0.207, 95%CL (0.117, 0.297)] compared to the left-paddlers group [*M* = 0.071, 95%CL (−0.019, 0.162)], *p* < 0.05. The analysis also indicated a significant interaction effect between paddlers and electrode pair in the α band [*F*_(2, 82)_ = 3.639, *p* < 0.05, *η*^2^ = 0.142]. Simple main effect analysis indicated that a set of electrode pair AI was a significant difference between paddlers [*F*_(1, 41)_ = 5.385, *p* < 0.05]. The left-paddlers group exhibited significantly positive AI of T8–T7 [*M* = 0.116, 95%CL (−0.052, 0.284)] compared to the right-paddlers group [*M* = 0.292, 95%CL (0.124, 0.460)], *p* < 0.05.

2(paddlers) × 2(electrode pairs) mixed design analysis of variance was conducted for the parietal asymmetry. The results showed that there was no interaction effect of the parietal AI between the left and right paddlers in the θ band [*F*_(1, 41)_ = 1.412, *p* = 0.278, *η*^2^ = 0.053], and the main effects of the two factors were not significant. In the α band, the parietal AI had no significant interaction between the left paddlers and right paddlers [*F*_(1, 41)_ = 1.339, *p* = 0.250, *η*^2^ = 0.060], and the main effects of the two factors were not significant.

One-way ANOVA for the occipital asymmetry yielded no interaction effect in the θ [*F*_(1, 41)_ = 1.412, *p* = 0.247, *η*^2^ = 0.060] and α [*F*_(1, 41)_ = 3.084, *p* = 0.093, *η*^2^ = 0.123] band in the paddlers right and left paddlers between two groups.

## 4. Discussion

The purpose of this study was to explore the differences in post-exercise brain activity between the left and right paddlers. The results showed that there was more cortical activation of the θ and α frequency bands in the left-paddlers group than in the right-paddlers group, and brain lateralization in the two groups of paddlers occurred mainly in the ipsilateral hemisphere.

Power spectrum analysis of EEG signals is a well-established method for quantifying higher cortical functional activity [23,24]. The power spectrum value varies according to alterations in brain state. The α band is one of the most significant frequency bands in the brain and is linked to alertness, mood, and motivational regulation [25]. Furthermore, this frequency range is also involved in inhibitory processes in attention and has been shown to be positively correlated with the speed of information processing [26,27]. The increase in the α band power spectrum value after exercise is indicative of an increase in the synchronization of α band-dominated cortical neuronal activity. This is likely due to the increased intensity of cortical neuronal metabolism. As working memory load in the cortex increases, the α rhythm also increases to reduce extraneous interference from external stimuli [28]. Dragon boat racing requires strong coordination and control of the paddling position, explosive force, and endurance qualities of the body, while also working effectively as a team—all of which necessitates a high level of concentration. This study showed that the α band power spectrum values were significantly higher in the left paddlers after exercise, compared to the right paddlers. These results suggest that long-term systematic dragon boat training can improve intrinsic concentration and good control of body posture.

The θ frequency is associated with higher cortical functions such as working memory, executive control, and focused attention [29,30]. It has been reported by several studies that EEG θ activity increases as the need for cognition and mental load does [31,32]. Many studies have also confirmed the presence of central fatigue-related changes in the θ band [33,34]. Furthermore, the slow-wave component of the brain increases after high-intensity exercise [35,36]. This could be related to central fatigue or to a change in the sensory feedback from locally fatigued muscles, which would lead to higher demands for cortical processing of that feedback and thereby more θ activity. The central nervous system issues high-frequency impulses over a longer period, which leads to an increase in energy consumption of nerve cells in the cerebral cortex and faster fatigue. Afferent impulses from working muscles produce continuous, long-lasting activity on nerve cells, which leads to the reduced working capacity of nerve cells and increased slow-wave power values. This is consistent with the findings of our study. Compared to the right paddlers, the left paddlers may have to activate more central nervous system cells to maintain focus and optimize their body posture for rowing, thus expending more energy.

The use of the dominant hand can affect brain activity in unilateral sports [13,37]. The subjects in this study are all right-handed athletes who are accustomed to using their right hand (arm) in daily life and at work to perform fine motor and general power moments, and who have developed relatively stable motor execution and sensory pathways in their brains [3]. Nevertheless, for the left paddlers in the dragon boat, their left arm muscles receive more specialized strength movements in training and competition than their right arm muscles. This does not mean that their left hand is accustomed to performing fine motor tasks; the fact that they are right-handed still objectively exists. This study showed that the left paddlers had higher θ and α power spectrum values than the right paddlers, which provides evidence that the left paddlers may expend more energy in rowing. This was supported by our above assumption.

Many studies have also demonstrated the asymmetry in EEG between left- and right-handed subjects [37,38]. An existing hypothesis called “hemispheric coactivation” [4] was used to explain the phenomenon of ipsilateral cortical activation in the active hand, which assumes that the ipsilateral motor area contributes to the initiation response. Among the differences in asymmetry indices of the θ and α frequency bands, we found significant asymmetry in both hemispheres of the right- and left-paddler groups, specifically in the pattern of rightward asymmetry to the right-paddlers group and leftward asymmetry to the left-paddlers group. It is also observed in EEG [39,40] and brain imaging techniques [41,42] that ipsilateral motor areas are activated with increased neural activity. This phenomenon of ipsilateral functional aggregation may be related to the essential amount of remote axonal connections by functional grouping in the same hemisphere [43]. The time limitations on information transfer between the cerebral hemispheres via the corpus callosum, which would increase the need for local functional specialization in the hemispheres, allowing for better efficiency and parallelization of neurocognitive operations [44]. Thus, the ipsilateral cortical activation that occurs in the right and left paddlers may be the result of coordinated interaction between the two hemispheres and provide additional space for the encoding of more motor skill reserves in the dominant hand. At the same time, hemispheric asymmetry increases total neural ability and efficient implementation for better task performance.

In addition, this study showed that the right-paddlers group reflects more rightward asymmetry in the right frontal and central regions compared to the left-paddlers group. Studies have shown that the length of the sulcus in the precentral gyrus of right-handed individuals exhibits a marked asymmetry of larger left and smaller right, with the dominant left motor cortex being larger [45,46]. This asymmetrical cortical area improves the efficiency of transmission between the two cerebral hemispheres, thus avoiding excessive conduction delay [47]. With distinction to handedness in general, the professional dragon boat athletes involved in this research are all right-handed and have received long-term training in a particular discipline. The right paddlers therefore play to their right-handedness advantage over the left paddlers, which may also be an influential factor in increasing asymmetric activation within the cortex of the right paddlers.

## 5. Conclusions

In this study, we found significant differences in EEG spectrum values between right and left paddlers after exercise. This is manifested in the left paddlers having higher spectrum values than the right paddlers, and by the fact that the brain lateralization of the left and right paddlers occurs mainly in the ipsilateral hemisphere. We suggest that unilateral-sport training and handedness are the key factors influencing post-exercise brain activity between the left and right paddlers. While motor cortical dominance has been classically associated with the contralateral hand, this study validates another hemisphere asymmetry of ipsilateral motor cortical activation. In the future, we will further explore the stability and accuracy of the results and optimize experimental protocols, such as the use of dry electrodes to detect dynamic patterns in EEG signals during exercise.

## Figures and Tables

**Figure 1 brainsci-12-01621-f001:**
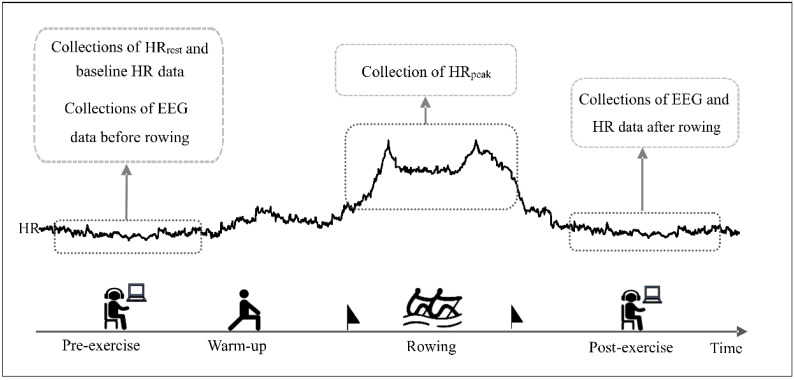
Experimental protocol. HR: heart rate.

**Figure 2 brainsci-12-01621-f002:**
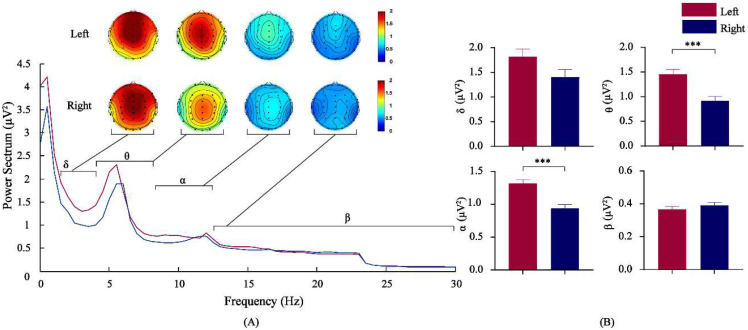
Comparison of the EEG power spectrum parameters between the two groups after rowing. (**A**) frequency curve and EEG energy map of significantly different bands; (**B**) power spectrum values of each band. Left: left-paddlers group, Right: right-paddlers group, EEG: electroencephalogram, *** *p* < 0.01.

**Figure 3 brainsci-12-01621-f003:**
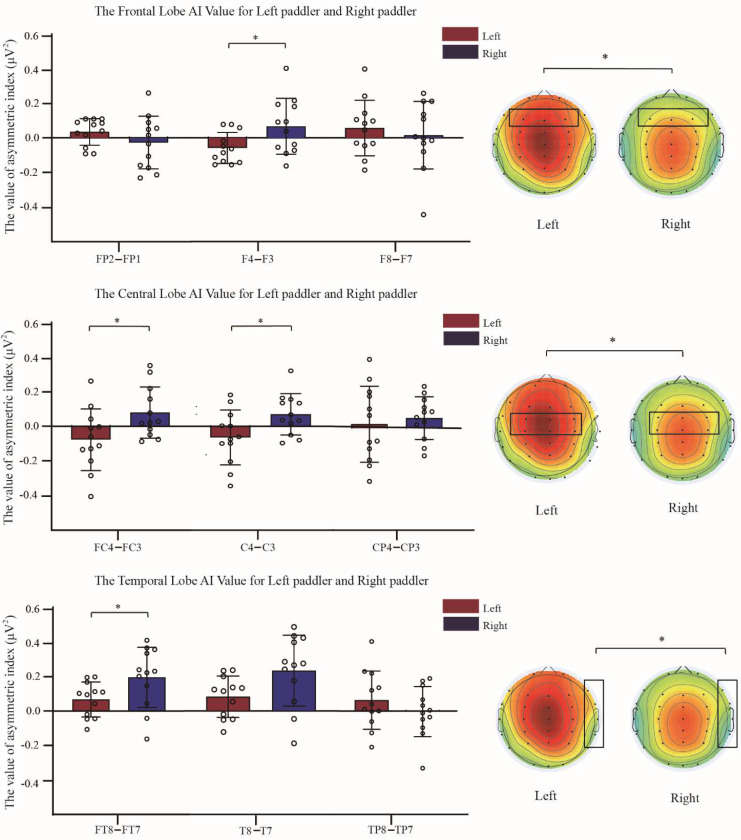
Brain area Asymmetric index (AI) in θ band between the two groups. The AI of the right−paddlers group is significantly greater than that of the left−paddlers group after rowing. The brain energy image displays the distinct in left paddlers and right paddlers (black box). Error bars represent the SD. Asterisks indicate significant differences between conditions (* *p* < 0.05). Left: left−paddlers group, Right: right−paddlers group.

**Figure 4 brainsci-12-01621-f004:**
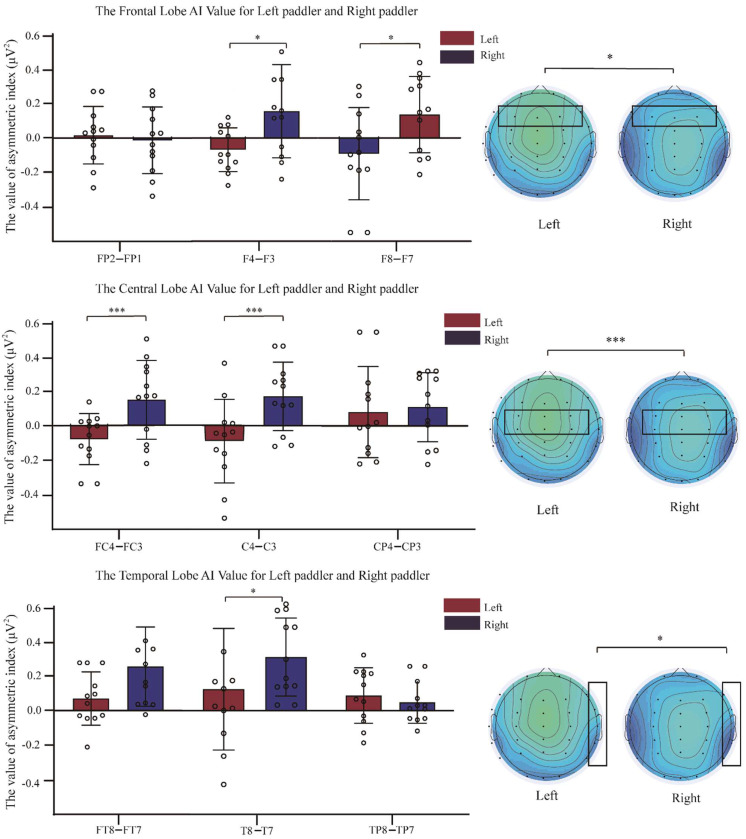
Brain area Asymmetric index (AI) in α band between the two groups. The AI of the right−paddlers group is significantly greater than that of the left−paddlers group after rowing. The brain energy image displays the distinct in left paddlers and right paddlers (black box). Error bars represent the SD. Asterisks indicate significant differences between conditions (* *p* < 0.05, *** *p* < 0.05). Left: left−paddlers group, Right: right-paddlers group.

**Table 1 brainsci-12-01621-t001:** Demographic information of participants ($\bar{x}$ ± *s, n* = 43).

Variable	Left-Paddlers (*n* = 21)	Right-Paddlers (*n* = 22)
Age (years)	20.08 ± 1.12	20.62 ± 0.65
Height (cm)	183.54 ± 4.74	182.62 ± 6.49
Weight (kg)	83.23 ± 8.25	84.77 ± 10.32
BMI (kg/m^2^)	25.33 ± 1.65	24.67 ± 1.69
Training age (years)	4.92 ± 2.49	4.31 ± 1.98

Note: BMI: Body Mass Index. There were no significant differences between left paddlers and right paddlers in height, weight, BMI and training years (unpaired *t*-test).

**Table 2 brainsci-12-01621-t002:** 1000 m dragon boat ergometer exercise performance and exercise performance ($\bar{x}$ ± *s, n* = 43).

Physiological Indexes	Left-Paddles (*n* = 21)	Right-Paddles (*n* = 22)	*p*-Value
HR peak (b/min)	190.77 ± 2.83	190.92 ± 2.92	0.99
Percentage of HR max (%)	95 ± 0.01	0.96 ± 0.01	1.00
Physiological load index	2.15 ± 0.05	2.17 ± 0.06	0.35
RPE > 18	20	20	1.00
Exercise duration (min)	4.81 ± 0.44	5.02 ± 0.51	0.41

Note: Percentage of HR max (%) = HR peak/HR max, >90%: super-intensity exercise, 80~90%: high-intensity exercise, 60~80%: moderate-intensity exercise, <60% small-intensity exercise; Physiological load index = (HR peak + HR rest)/2/HR rest, >1.4: low degree, 1.4~1.6: middle degree, <1.6: high degree.

## Data Availability

The original contributions presented in the study are included in the article, further inquiries can be directed to the corresponding author.

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
