# Peer review of "Comparison of Electroencephalogram Power Spectrum Characteristics of Left and Right Dragon Boat Athletes after 1 km of Rowing"

_brainsci, 2022, doi:10.3390/brainsci12121621_

Round 1

Reviewer 1 Report

General comments

            The authors have written an interesting paper with the aim to examine the differences in brain activation between the two types of dragon-boat paddlers. The authors used data from resting state EEG signals before and after the exercise. The  Asymmetry index was used to analyze differences in hemisphere activation between left paddlers and right paddlers. However, there are some major issues with the paper.

           There are a number of typing errors throughout, and the references in the text are not in accordance with the journal norms (i.e. Muellbacher W et al; Taddei F et al. ; McLean JM; Ciurczak FM). The data collection including pre and post-exercise is not clearly justified and the approach is not well presented. The author's claim related to the advance of the knowledge due to data from the current study is based on the assumption that the findings could provide "referential evidence for a deeper understanding of the neurological functional mechanisms of athletes in unilateral sports". Nevertheless, it could be questioned why not investigating the cortical activity during the exercise. The reasons for the option for investigating pre and post-exercise should be clearly presented.  It is hard to support this option without clear support.

Specific comments 

Introduction

            Could you please rephrase this sentence for clarity? ", especially the synchronization of the movements of their movements" - what does "the synchronization of the movements of their movements" really mean? (line 32)

           Could you please elucidate what " somatic movements" mean? are you talking about voluntary movements? please, rephrase it for clarity. (line 34)

           Could you please elucidate what " somatic muscles" mean? please, rephrase it for clarity. (line 35).

Methods

            Figure 1 should be corrected as the illustration does not show the EEG data collection before the exercise test.

            Lines 122-124) Could you please present clearer information on this procedure? could one assume that participants were instructed to wash their heads pre and post-exercise? this information is not clear to readers.

      Were participants instructed to maintain their eyes closed during the 6 minutes (pre and post-exercise) EEG data acquisition? if so, why? why not compare eyes closed and eyes opened and filtering the artifacts?

Results

            Table 2

            How did you estimate the HR max?

            What does "Physiological load index" mean? how did you calculate it?

            RPE: could you please describe the methods used to require athletes to provide the perceived effort? 

Discussion

            Interestingly, the almost studies cited in the discussion investigated asymmetries and cortical activation during the task.  The discussion should address the effect of the exercise on asymmetry and cortical activation, as this was the approach used in the research. The discussion adds little to this topic and readers could have difficulty understanding what the data presented really add to the topic.  

Reviewer 2 Report

1)    I think that the aim of the study should be explained in a clearer way, as it is in the abstract and in the discussion, the scientific question behind remains elusive.

2)    The logic in the introduction should be improved, especially between line 53 and 73, where multiple references are contradictory (eg. ref 6 and 13). As it is presented here the result of ref 11 seems quite obvious. Actually, the conclusion (line 71) looks like an imprecise shortcut. I think that the reader should understand what are the conclusions of these studies and if, or not, they converge to the same final understanding of brain physiology. This study would gain in explaining clearly what is the hypothesis tested here.

3)    Line 48 : The sentence beginning by “whereas” is cut in half.

4)    Collection of EEG data should also be indicated on fig 1 in the pre-exercise period.

5)    Line 201, 242, 256, 276, 288 : these informations should go in the methods section

6)    Please, add a new column in table 2 with p value and the test used there, even if it is NS it should be easily accessible in the table.

7)    Not sure that three stars refer normally to p<0.05. This way of presenting statistics result is misleading.

8)    What is the interest of table 3, additionally to figures 3 and 4? May it be presented in a supplementary file?

9)    Table 3: stars on the left paddle group is confusing. Please add another column for statistics.

10)  In the discussion, I had the feeling that many points were out of the scope of this study.

11) The fact that you enrolled only right-handed athletes should be highlighted in the discussion.

Round 2

Reviewer 1 Report

General comments

 As I mentioned in the first version of this manuscript, the authors have written an interesting paper with the aim to examine the differences in brain activation between the two types of dragon-boat paddlers. The authors did a good job and the paper in this present form has improved its quality and clarity. However, there are some minor issues with the paper that still should be managed. Despite the effort of the authors, there are still several typing errors throughout. These errors compromise the understanding of the paper and limit its acceptance. I suggest a revision of a native English reviewer.

Concerning specific comments, I listed below the main ones.

Abstract: Could you please elucidate what "both groups of athletes existed a broader pattern" means? In fact, I did not understand what "existed" means.

Pg 1, lines 33-34 - "It is well known that cortical control of voluntary movements is crossed, with one side of the motor cortex dominating the activity of the other side of the muscle". Could you please replace "other side of the muscle" with "other side of the body" for clarity?

pg 2, lines 50-54: I suggest changes in this paragraph, for clarity. Rather than "It is well known that for unilateral sports, such as throwing, tennis, badminton and table tennis, there are asymmetries in the brain regions responsible for motor command delivery and sensory reception in the athletes of these sports. For dragon boat sport, in particular, where the left and right paddlers are fixed and play a specialized sporting role, asymmetries in their post-exercise EEG signals may be present. However, little has been reported on the above hypothesis";

"It is well known that for athletes involved in unilateral sports, such as throwing, tennis, badminton and table tennis, there are asymmetries in the brain regions responsible for motor command delivery and sensory processing (PLEASE, INCLUDE AT LEAST TWO REFERENCES HERE). For dragon boat sport, the left and right paddlers are fixed and play a specialized sporting role. Due to this inherent dragon boat sport’ characteristic, asymmetries in their post-exercise EEG signals might be present. However, evidence for this hypothesis is lacking.

pg 2, lines 65-67: What do you mean by "posterior tDCS in the following sentence? "in another study of posterior transcranial direct current stimulation (tDCS), it was confirmed that the lateralization of left vestibular cortical function was more pronounced in the left-handed [15]". Could you please elucidate it? what "´posterior" means.

pg 2, lines 73: what does "on them" mean in the following sentence? " to conduct a 1km rowing test on them" 

pg 2, line 75: could you please replace "their exercise" with "the 1km rowing test"? (before and after the 1km rowing test"

pg 2, lines 85-86: I did not understand what this "standard" means? “right-handedness (measured by the Chinese translated version of Edinburgh inventory [16 ]. Were all selected athletes right-handed? This information, at this moment, is not yet clear to readers. Could you please insert a brief mention of it, explaining the choice? (I mean, why not recruit left-handedness?). Just for clarity.

Pg 5, line 198: the sentence “The results revealed that no significant interaction effect” should be “The results revealed no significant interaction effect”

Discussion: Could you please provide a deeper rationale to explain the results? I was wondering whether some information on possible mechanisms could be added to the text to improve the quality of the discussion while suggesting future perspectives.

Indeed. The issue of bringing the relationship between the θ and α frequency bands and fatigue state is out of context. The current design (1min all out) is not appropriate for inferring about “brain signatures” related to fatigue. Moreover, as the references used in the text show, this possible association could be more related to “mental fatigue” or “cognitive fatigue” rather than to physical fatigue during a short high-intensity exercise. The argument in lines 310-315 is hard to support in this present form. The paper warrants an improved discussion section.  
